# Brain Atrophy Mediates the Relationship between Misfolded Proteins Deposition and Cognitive Impairment in Parkinson’s Disease

**DOI:** 10.3390/jpm11080702

**Published:** 2021-07-23

**Authors:** Chiun-Chieh Yu, Chia-Yin Lu, Meng-Hsiang Chen, Yueh-Sheng Chen, Cheng-Hsien Lu, Yi-Yun Lin, Kun-Hsien Chou, Wei-Che Lin

**Affiliations:** 1Department of Diagnostic Radiology, Kaohsiung Chang Gung Memorial Hospital, College of Medicine, Chang Gung University, 123 Ta-Pei Road, Niao-Sung, Kaohsiung 833, Taiwan; yuchiunchieh@gmail.com (C.-C.Y.); vickyandlily@gmail.com (C.-Y.L.); sperfect@msn.com (M.-H.C.); yssamchen@gmail.com (Y.-S.C.); 2Department of Neurology, Kaohsiung Chang Gung Memorial Hospital, College of Medicine, Chang Gung University, 123 Ta-Pei Road, Niao-Sung, Kaohsiung 833, Taiwan; chlu99@cgmh.org.tw; 3Philosophy in Nursing, School of Nursing, Shu Zen College of Medicine and Management, Number 452, Hwan-Chio Rd, Luju Dist, Kaohsiung 821, Taiwan; yrc123@ms.szmc.edu.tw; 4Brain Research Center, National Yang Ming Chiao Tung University, Taipei 112, Taiwan; dargonchow@gmail.com; 5Institute of Neuroscience, National Yang Ming Chiao Tung University, Taipei 112, Taiwan

**Keywords:** cognition, dementia, gray matter atrophy, misfolding protein, Parkinson’s disease

## Abstract

Parkinson’s disease is associated with cognitive decline, misfolded protein deposition and brain atrophy. We herein hypothesized that structural abnormalities may be mediators between plasma misfolded proteins and cognitive functions. Neuropsychological assessments including five domains (attention, executive, speech and language, memory and visuospatial functions), ultra-sensitive immunomagnetic reduction-based immunoassay (IMR) measured misfolded protein levels (phosphorylated-Tau, Amyloidβ-42 and 40, α-synuclein and neurofilament light chain) and auto-segmented brain volumetry using FreeSurfur were performed for 54 Parkinson’s disease (PD) patients and 37 normal participants. Our results revealed that PD patients have higher plasma misfolded protein levels. Phosphorylated-Tau (p-Tau) and Amyloidβ-42 (Aβ-42) were correlated with atrophy of bilateral cerebellum, right caudate nucleus, and right accumbens area (RAA). In mediation analysis, RAA atrophy completely mediated the relationship between p-Tau and digit symbol coding (DSC). RAA and bilateral cerebellar cortex atrophy partially mediated the Aβ-42 and executive function (DSC and abstract thinking) relationship. Our study concluded that, in PD, p-Tau deposition adversely impacts DSC by causing RAA atrophy. Aβ-42 deposition adversely impacts executive functions by causing RAA and bilateral cerebellum atrophy.

## 1. Introduction

Parkinson’s disease (PD) is currently the second most common neurodegenerative disease [1], lacking both efficient methods of early diagnosis and a mechanism-based cure [2]. In addition to motor deficits in PD patients, cognitive decline is increasingly emphasized and investigated. Individuals with PD have increased risk of dementia compared to the general population [3]. Such cognitive impairments in PD patients have significant impacts on quality of life and provide daily burdens for caregivers [4]. Detailed neuropsychological assessments are lengthy and cost time, while also being heavily dependent on the adherence of patients. The lack of a long-term follow-up regarding cognitive function, early medical or psychological interventions remains challenging and potentially affects the prognosis. Previous studies have demonstrated that plasma misfolded proteins are correlated with cognitive decline and are potential biomarkers for the detection and prediction of cognitive impairment [5].

Compelling evidence suggests that misfolded protein aggregation plays an important role in cellular dysfunction and tissue damage, leading to development of PD [6]. Less invasive biomarkers moving from cerebrospinal fluid (CSF) to blood are a noted clinical trend and may act as a potential large-scale screening tool in longitudinal follow-up. Among them, α-synuclein (α-syn), total and p-Tau, Aβ-40 and 42, and neurofilament light chain (NfL) are most commonly investigated and may contribute to brain atrophy and cognitive impairment [7].

In this study, we aim to clarify the associations between misfolded protein deposition, brain atrophy, and cognitive impairment. Our hypotheses are as follows: (1) PD patients have higher neurotoxic plasma misfolded proteins than normal participants; (2) misfolded proteins are correlated with brain atrophy and cognitive decline; and (3) brain atrophy mediates the relationship between misfolded protein deposition and cognitive decline.

## 2. Materials and Methods

### 2.1. Subjects

In our study, 54 PD patients and 37 sex- and age-matched healthy volunteers were enrolled in the Neurology Department of Chang Gung Memorial Hospital. The diagnosis of PD was clinically established by a qualified neurologist in accordance with the UK Parkinson’s disease society brain bank clinical diagnostic criteria [8]. Healthy volunteers without signs and symptoms of neurological disease, alcohol or substance abuse, psychiatric illness, nor head injury were enrolled as the control group.

Age, sex, and education levels were obtained for all participants. Disease duration, disease severity, and functional assessments were obtained for PD patients. Disease severity were assessed by the modified Hoehn and Yahr staging scale (HY-stage) [9] and Unified Parkinson’s Disease Rating Scale (UPDRS) [10]. Three areas were assessed in our study including: (1) nonmotor experiences of daily living (13 items); (2) motor experiences of daily living (13 items); and (3) motor examination (18 items). Functional assessment was performed with the Schwab and England ADL Scale [11].

### 2.2. Neurobehavioral Evaluation and Cognitive Severity Definition

A clinical psychologist blinded to each participant’s status performed neuropsychological tests focusing on executive, speech and language, memory, and visuospatial functions. The Chinese versions of the Cognitive Abilities Screening Instrument (CASI) [12], the Wechsler Adult Intelligence Scale-III (WAIS-III) [13], and the Mini-Mental State Examination (MMSE) were tested to assess the five psychological domains [14]. CASI analyzes 9 domains of cognitive functions, including: (1) long-term memory, (2) short-term memory, (3) attention, (4) mental manipulation, (5) orientation, (6) abstract thinking (ABS), (7) language, (8) drawing, and (9) animal naming fluency. In addition, WAIS-III provides scores for full-scale IQ, including verbal IQ and performance IQ. Verbal IQ and performance IQ further include two components each (verbal IQ: verbal comprehension index, working memory index; performance IQ: perceptual organizing index, processing speed index). There are a total of twelve subtests: (1) vocabulary, (2) similarities, (3) information, (4) comprehension, (5) arithmetic, (6) digit span, (7) letter number sequencing, (8) picture completion, (9) block design, (10) matrix reasoning, (11) digit symbol coding (DSC), (12) symbol search.

### 2.3. Blood Sampling and Assaying of Plasma Biomarkers: T-tau, Aβ-40, Aβ-42, α-Synuclein, and Neurofilament Light Chain

Blood samples were collected at a fixed time interval (10:00–11:30 AM) to avoid the sleep–wake cycles in plasma biomarkers. Blood was not collected within 2 h post exercise. We used 10 mL K3-EDTA tubes (Greiner Bio-One 455036) to contain the blood samples, followed by gently inverting the tubes 10 times immediately after blood draw. The centrifugation was conducted at 15–25°C and 1500–2500 g for 15 min by using a swing-out (bracket) rotor.

In total, 0.5 mL of plasma (supernatant) was transferred into a fresh 1.5 mL Eppendorf tube and frozen at −80°C within 3 h after blood draw. Details of the methodology are in our previous study [15]. Five IMR kits were used to separately assay concentrations of p-tau (MF-PT1-0060, MagQu), Aβ-40 (MF-AB0-0060, MagQu), Aβ-42 (MF-AB2-0060, MagQu), α-syn (MF-ASC-0060, MagQu), and NfL (MF-NFL-0060, MagQu) in human plasma. For assaying Aβ-40, α-syn, and p-tau, 80 µl reagent was mixed with 40 µl plasma. For assaying Aβ-42 and NfL, 60 µl reagent was mixed with 60 µl plasma. An IMR analyzer (XacProS) was used for transformation from IMR signal to biomarker concentrations [16]. For each biomarker assay, duplicated measurements were performed to obtain an averaged value, which represents the concentration of a biomarker.

### 2.4. Structural MRI Imaging

#### 2.4.1. Image Acquisition

Each participant’s head was immobilized to diminish motion artifacts. MR images were acquired using a 3.0 T whole body GE Signa MRI system (General Electric Healthcare). A 3D-FSPGR sequence (repetition time (TR) = 9.492 ms, echo time (TE) = 3.888 ms, flip angle = 20°, field of view (FOV) = 24 cm × 24 cm, matrix size = 512 × 512, 110 continuous slices with a slice thickness of 1.3 mm and in-plane spatial resolution of 0.47 mm × 0.47 mm) was used for acquiring T1-weighted structured images through the whole head parallel to the anterior–posterior commissure.

#### 2.4.2. Image Data Processing

Auto-segmentation of ROIs (region of interests) was carried out using FreeSurfer image analysis suite and each brain area was divided by estimated total intracranial volumes (eICVs). FreeSurfer is a powerful tool providing extensive and automated analysis of ROIs in human brains, freely available for download online (http://surfer.nmr.mgh.harvard.edu/, v7.1.0, accessed on 11 May 2020). It is based on a subject-independent probabilistic brain atlas in combination with nonlinear image registration of individual images to obtain subject-specific measurements. FreeSurfer morphometric procedures show good test–retest reliability across scanner manufacturers and across field strengths. The technical details of these procedures are described in prior publications [17].

### 2.5. Statistics

#### 2.5.1. Demographic Data

IBM SPSS Statistics version 20 (IBM, Armonk, NY, USA) was utilized for statistical analysis. All demographic data were compared between the normal controls (NCs) and the PD group. Sex was compared with Pearson’s Chi Square test. Age and education levels were analyzed with 2-sample independent *t*-test. MMSE and neuropsychological assessments were analyzed using ANCOVA (Analysis of Covariance) with age, sex, and education covariates. Plasma biomarkers and ROI volumes were compared with ANCOVA after controlling for age and sex. All results are reported as the mean ± standard deviation (SD). For each ROI’s volume, the left and right brain area volumes were analyzed, respectively, after correction via dividing by each intracranial volume. The Benjamini–Hochberg adjusted *p* value (Q value) < 0.05 was considered significant.

#### 2.5.2. Partial Correlation

The variables with significant differences between the NC and PD groups were enrolled into the analysis of partial correlation, which was used to identify the relationship between plasma biomarkers, brain volumes, and cognitive functions. Plasma biomarkers included α-syn, Aβ-40 and 42, and NfL. Brain ROIs included bilateral cerebellum, thalamus, caudate, putamen, hippocampus, amygdala, nucleus accumbens, and brainstem. Neuropsychological assessments included digit symbol coding, arithmetic, abstract thinking, information, semantic fluency, picture completion, block design and drawing. Bonferroni adjusted *p* < 0.05 was considered significant. The correlation coefficient was also reported.

#### 2.5.3. Mediation Analysis

In this mediation analysis, we aimed to investigate whether the effect of the plasma misfolded proteins (independent variable) on the neuropsychological assessments (dependent variable) was explained indirectly by the brain volume atrophy (mediator) with a significant group main effect. An indirect dialogue PROCESS v3.5 (https://www.processmacro.org/download.html, accessed on 2 May 2020) was downloaded into IBM SPSS Statistics v.20 for bootstrap test (statistical significance with 5000 bootstrap samples was used).

The path model examined three effects of interest required if brain atrophy links misfolded proteins with neuropsychological assessments:(a)The effect of plasma misfolded protein level on brain volume (indirect effect, path a);(b)The effect of brain volume on neuropsychological assessments by controlling the effect of plasma misfolded proteins (indirect effect, path b);(c)The mediation effects a × b are defined as the reduction in the relationship between the plasma misfolded proteins and the neuropsychological assessments (total relationship, path c) by including the brain volume in the model (direct path, path c).

Statistical significance threshold was set at 0.05 for all the relevant paths.

## 3. Results

### 3.1. Basic Characteristics

The baseline clinical demographics, neuropsychological assessments, plasma misfolded protein levels, and brain volumes of all participants are listed in Table 1. Of the 54 PD patients and 37 normal controls, statistical analysis of clinical characteristics showed no significant difference between age and sex (age: *p* = 0.093, sex: *p* = 0.199). The mean of disease duration of the PD group was 4.97 years. The NC group performed better in MMSE than PD patients (*p* < 0.001).

### 3.2. Neuropsychological Assessments

WAIS-III and CASI neuropsychological assessments were performed in our study (Table 2) and were applied to the five cognitive domains. The NC group performed better in four of the domains (execution, memory, speech and language, visuospatial). In the executive functions, DSC (*p* < 0.001), arithmetic (*p* = 0.008), and abstract thinking (*p* < 0.001) were significant. In the memory functions, only information (*p* = 0.012) was significant. In speech and language, comprehension (*p* = 0.001) and semantic fluency (*p* = 0.025) were significant. In visuospatial functions, picture completion (*p* = 0.001), block design (*p* < 0.001), and drawing (*p* = 0.037) were significant.

### 3.3. Plasma Biomarkers

Four plasma biomarkers were noted as significantly higher in the PD group, including p-Tau (*p* = 0.0012), Aβ-42 (*p* = 0.0004), α-syn (*p* = 0.0390), and NfL (*p* = 0.0016). The lower Aβ-40 level in the PD group (*p* = 0.0520) indicates that less neuroprotective proteins are present in PD patients (Figure 1). The ROC curve analysis of plasma misfolded proteins was performed and attached in the Appendix A.

### 3.4. Brain Volume Analysis

Our results showed significant brain atrophy of PD patients. The ROI analyses revealed significant atrophy of bilateral cerebellum (L: *p* < 0.001; R: *p* < 0.001), thalamus (L: *p* = 0.005; R: *p* = 0.030), caudate (L: *p* = 0.004; R: *p* = 0.006), putamen (L: *p* = 0.004; R: *p* < 0.001), hippocampus (L: *p* = 0.004; R: *p* = 0.002), amygdala (L: *p* = 0.004; R: *p* = 0.002), accumbens areas (L: *p* = 0.008; R: *p* < 0.001), and brainstem (*p* = 0.038) in PD patients.

### 3.5. Correlation Analysis of Plasma Misfolded Proteins, Neuropsychological Assessments, and Brain Atrophy

#### 3.5.1. Misfolded Proteins and ROI Volumes in PD Patients

The correlation between plasma biomarkers and ROI volumes after controlling age, sex, and education level were examined (Table 3). Plasma p-Tau was found to be negatively correlated with the bilateral cerebellar cortex (left: r = −0.311, *p* = 0.005; right: r = −0.320, *p* = 0.004), right caudate nucleus (r = −0.342, *p* = 0.002), and RAA (r = −0.321, *p* = 0.004). Plasma Aβ-42 was negatively correlated with the bilateral cerebellar cortex (left: r = −0.353, *p* = 0.001; right: r = −0.365, *p* = 0.001), bilateral caudate nucleus (left: r = −0.397, *p* < 0.001; right: r = −0.422, p < 0.001), and RAA (r = −0.301, *p* = 0.006).

#### 3.5.2. Neuropsychological Assessments and Brain Volumes in PD Patients

We examined the correlation between MMSE score, neuropsychological assessments, and ROI volumes after controlling for age, sex, and education (Table 3). MMSE was positively correlated with the bilateral cerebellar cortex (left: r = 0.413, *p* < 0.001; right: r = 0.362, *p* < 0.001), left caudate nucleus (r = 0.290, *p* < 0.001), and RAA (r = 0.409, *p* < 0.001). Arithmetic was positively correlated with the bilateral cerebellar cortex (left: r = 0.416, *p* < 0.001; right: r = 0.362, *p* = 0.001). DSC was positively correlated with the left cerebellar cortex (r = 0.320, *p* = 0.002), and RAA (r = 0.318, *p* = 0.003). Abstract thinking was positively correlated with the bilateral cerebellar cortex (left: r = 0.291, *p* = 0.006; right: r = 0.308, *p* = 0.004) and RAA (r = 0.296, *p* = 0.005).

### 3.6. Mediation Analysis

Brain Mediators of Plasma Misfolded Proteins on Cognitive Functions
p-Tau–digit symbol coding relationship mediator

Single-level three-variable mediation analysis revealed atrophy of RAA negatively and completely mediated the p-Tau–DSC relationship (Coef_a_ = −0.318, P_a_ = 0.002; Coef_b_ = 0.329, P_b_ = 0.002; Coef_ab_ = −0.193, P_ab_ = 0.07) (Figure 2A). The results indicate that, with regard to the effects of RAA atrophy on DSC, the effects from p-Tau to DSC disappeared, revealing that it is RAA that causes the poor performance of DSC. A bootstrap test revealed the significance (effect = −0.836, CI: −0.1669; −0.027) of the path analysis (Table 4).
Aβ42–digit symbol coding relationship mediator

The RAA volume was negatively associated with the Aβ-42 level and predicted impaired DSC. Specifically, RAA partially mediated the Aβ42–DSC relationship (Coef_a_ = −0.361, P_a_ < 0.001; Coef_b_ = 0.329, P_b_ = 0.002; Coef_ab_ = −0.243, P_ab_ = 0.023) (Figure 2B). This finding suggests that plasma Aβ-42 contributes to the negative impact on DSC via RAA atrophy. The bootstrap test achieved significance (effect = −0.3649, CI: −0.7552; −0.1123).

The left cerebellar cortex volume was negatively associated with the Aβ-42 level and predicted impaired DSC. In other words, left cerebellar atrophy partially mediated the Aβ-42–DSC relationship (Coef_a_ = −0.317, P_a_ = 0.002; Coef_b_ = 0.326, P_b_ =0.002; Coef_ab_ = −0.249, P_ab_ = 0.019) (Figure 2C). This finding suggests that plasma Aβ-42 plays a role in contributing to a negative impact on DSC through left cerebellar cortex atrophy (Table 4). The bootstrap test achieved significance (effect = −0.3391, CI: −0.7551; −0.0822).
Aβ-42–abstract thinking relationship mediator

The RAA volume was negatively associated with Aβ-42 level and predicted impaired abstract thinking (Coef_a_ = −0.361, P_a_ < 0.001; Coef_b_ = 0.322, P_b_ = 0.002; Coef_ab_ = −0.226, P_ab_ = 0.036), suggesting plasma Aβ-42 is a factor contributing to a negative impact on abstract thinking through the RAA atrophy (Figure 2D). In other words, RAA partially mediated the Aβ42–abstract thinking relationship. The bootstrap test also achieved significance (effect = −0.1866, CI: −0.3989; −0.524).

Bilateral cerebellar cortex partially negatively mediated the Aβ42–abstract thinking relationship (right: Coef_ab_ = −0.310, P_ab_ = 0.003; left: Coef_ab_ = −0.310, P_ab_ = 0.003) (Table 4). The bootstrap test was significant (right: effect = −0.1663, CI: −0.3712; −0.0278; left: effect = −0.1538, CI: −0.3444;−0.267) (Figure 2E,F).

## 4. Discussion

Consistent with our hypothesis, PD patients experienced brain atrophy (bilateral cerebellar cortex, caudate nuclei, putamen, hippocampus, amygdala, RAA, and brainstem) as well as elevated misfolded proteins (p-Tau, Aβ-42, α-syn, NfL). The patients achieved lower scores in the neuropsychological assessments, including information, comprehension, arithmetic, semantic fluency, picture completion, block design, DSC, abstract thinking, and drawing. Furthermore, we determined that elevated plasma misfolded proteins (p-Tau and Aβ-42) are correlated with lower scores on neuropsychological assessments (MMSE, arithmetic, DSC and ABS) and brain atrophy (RAA, cerebellum and caudate nuclei), indicating the effects of misfolded proteins on neurotoxicity, neuron damage, and brain volume loss. Further mediation analysis revealed that RAA completely mediates the p-Tau–DSC relationship. Meanwhile, RAA and the bilateral cerebellum partially mediates the Aβ42–DSC and Aβ42–ABS relationships.

### 4.1. Pathophysiology of Misfolded Protein Deposition and Cortical Atrophy in PD Patients

The concentration of the plasma misfolded protein level is similarly and correlated to CSF level in Tau protein [18], Aβ-40 and Aβ-42 [19] in neurodegenerative disease. Plasma levels of misfolded proteins can be used as biomarkers to predictive cognitive impairment in PD [5]. Our results showed significantly higher plasma p-Tau in PD patients, which correlated with atrophy of the bilateral cerebellar cortex, right caudate nucleus, and RAA. In a previous study, Tau proteins became insoluble neurofibrillary tangles, deposited in several brain areas, including the frontal cortex [20], temporal cortex, striatum [21], accumbens area [22], putamen, caudate, and the dentate nucleus of cerebellum [21], with a notable association with cognitive decline and neuronal damage [23].

In addition, misfolded amyloid-β proteins gradually turn into insoluble neurotoxic amyloid plaques [24], causing synapse dysfunction, and promote tau phosphorylation into neurofibrillary tangles [25]. Pathologic studies have also revealed extensive Aβ deposits in the frontal lobe, leptomeningeal vessels, and accumbens area [26]. In this study, elevated Aβ-42 is correlated with atrophy of the bilateral cerebellar cortex, bilateral caudate nucleus, and RAA, indicating that Aβ-42 may have been deposited in those brain areas, causing atrophy. On the other hand, a negative correlation with Aβ-42 and Aβ-40 was observed in our study, which may be interpreted as increased deposition of Aβ-42 reducing the level of neuroprotective Aβ-40 [27].

### 4.2. Brain Atrophy Involved in Several Neuronal Circuits in Cognitively Impaired PD Patients

Cortical thinning and gray matter atrophy are associated with cognitive decline in PDD patients [28]. The caudate, putamen, globus pallidum, accumbens area, and thalamus are related to the frontal cortex through basal ganglia–thalamocortical circuits [29]. The nucleus accumbens serves as a motivation-to-movement interface by integrating information from the limbic drive to prefrontal cortex motor planning, based on glutamatergic afferents from the ventral hippocampus, basolateral amygdala, and the medial prefrontal cortex (PFC), and projects to the ventral pallidum. These brain regions are dedicated to executive functions [30]. Nucleus accumbens lesions cause disruptions in cognitive functions and affected processes [31]. Our results demonstrate that atrophy of the nucleus accumbens contributes to executive function impairment, including to DSC and ABS.

Another significant atrophic region is the cerebellum, which correlated with impaired executive functions (DSC and ABS). The cerebellum mediates cortical information processing via closed cortico-cerebellar loops, with strong connections to the lateral prefrontal cortex (PFC) [32]. As the PFC is connected with many different brain areas, PFC–cerebellar circuits are likely to be involved in a variety of processing activities, including cognitive domains such as memory or spatial perception [33].

### 4.3. Gray Matter Atrophy Contributions in Misfolded Proteins and Cognitive Impairment Correlation: Mediation Analysis

Previous studies have reported on correlations between elevated misfolded proteins and cognitive decline in PD patients. Cortical and subcortical atrophy has also been observed to be correlated with cognition impairments [34]. To our knowledge, this is the first study to investigate the mediation relationship between plasma misfolded proteins, brain structure abnormalities, and cognitive dysfunctions. Summarized and noted in Figure 3, misfolded protein depositions in RAA and bilateral cerebellum cortex cause underlying vulnerable brain gray matter atrophy, leading to subsequent circuit damage and executive dysfunction in PD patients.

Our results revealed that p-Tau and Aβ-42 are the primary proteins significant in the mediation of relationships. The mediation analysis revealed RAA atrophy completely mediates the higher plasma p-Tau and impaired DSC relationship. Neurofibrillary tangles deposit in the RAA and along the involved neuronal circuits, resulting in transportation dysfunction, neuronal death with volume loss, and pathway dysfunction [23]. In this study, we found elevated Aβ-42 correlated with atrophy in the RAA and executive functions (DSC and ABS) impairment. The possible clinical consequences of nucleus accumbens atrophy were recently suggested to include both neuropsychiatric and motor symptoms of PD due to the nucleus accumbens functioning as a “limbic–motor interface” and significantly decreasing in the volume in PD MCI patient [35]. Nucleus accumbens serves as an integrator of the mesolimbic dopaminergic circuit [36] and the atrophy of human nucleus accumbens in PD, called Mavridis atrophy (MA), which begins in early stage PD and is correlated with psychiatric symptoms [37]. Furthermore, reduced volumes of the nucleus accumbens are significantly associated with the performances of the attention, working memory and language domains in PD MCI patients [38].

In previous pathologic studies, Aβ-42 was also found in the accumbens area in PD patients and to be more aggravated in PDD patients [39]. The mediation analysis showed partial mediation of RAA atrophy in the Aβ-42–executive functions relationship. These data may indicate that elevated Aβ-42 as well as RAA atrophy together cause executive function impairments. The possible pathophysiology may involve deposition of Aβ-42 in RAA, which has a highly neurotoxic effect, resulting in neuronal loss and atrophy.

We further identified elevated Aβ-42 correlated with atrophy of the bilateral cerebellar cortex and executive functions (DSC and ABS) impairment. These data may indicate the deposition of Aβ-42 in the bilateral cerebellum having a highly neurotoxic effect, resulting in volume atrophy. The cerebellum is involved in movement and sensorimotor processing, cognitive control and executive function [40,41]. The interconnected circuit, namely the cortico-basal ganglia-cerebellar network [42], with involvement of prefrontal loops, thereby becomes dysfunctional. This pathway is associated with cognitive functions, connecting the prefrontal cortex with the dentate nucleus. Furthermore, the limbic domain of the basal ganglia, namely ventral striatum (including the nucleus accumbens), interconnected with the cerebellum, can be proved by functional neuroimaging [43].

Interestingly, α-syn, which is considered to be the hallmark of PD, was found to be insignificant in the mediation analysis. This finding may represent the direct effect of α-syn on cognitive impairment (data not shown). The onset and severity of dementia in Lewy body diseases is also associated with pathologies beyond changes in α-syn. Co-pathologies have been reported in postmortem studies, in vivo PET evaluations, and in animal models. Animal and cellular models suggest a synergistic relationship between α-syn and Aβ-42 and p-tau, which accelerates cognitive decline [44]. Postmortem studies show that pathological tau co-localizes to α-syn in Lewy bodies, further revealing that cognitive decline in PD patients is related to the combined influence of α-syn, Aβ-42, and p-tau [45]. The combination of α-syn with Aβ plaque and p-tau pathology is strongly correlated with dementia in PD patients [46,47]. In the present study, elevated p-Tau and Aβ-42 misfolded protein levels mediated with RAA brain atrophy and cognitive function impairment may indicate an underlying co-pathology of misfolded protein deposition with neurotoxic effects.

## 5. Limitations

There are several limitations of this study. First, the case number in our study is relatively small and limited to a single tertiary institution and thus may not be representative of the wider PD population; furthermore, only two cohorts were involved. Second, this is a cross-sectional study, so readers should thus interpret the results with caution, while future longitudinal studies are recommended. Third, plasma measurement is only an indirect index of cortical deposits [47]. The observed differences in peripheral blood may be due to multiple external factors, including the abundant plasma proteins, collection time, and storage conditions of the blood samples. Fourth, not all brain areas are discussed in our study; we only take deep gray matter regions into consideration. In addition, no grading of PD patients was classified in our study.

## 6. Conclusions

This study reveals that elevated plasma p-Tau and Aβ-42 may indicate RAA and bilateral cerebellum atrophy and executive function impairments via mediation analysis. We highlight the possibility that changes in some vulnerable anatomies, such as the RAA and bilateral cerebellum, may be correlated with elevated plasma p-Tau and Aβ-42 and the involvement of the underlying cortico-basal ganglia-cerebellar network, as seen in certain phenotypes of PD. This study offers valuable insights into the role of misfolded proteins in cognitive decline as associated with atrophy of certain brain regions and could provide novel targets for future neuroprotective therapies.

## Figures and Tables

**Figure 1 jpm-11-00702-f001:**
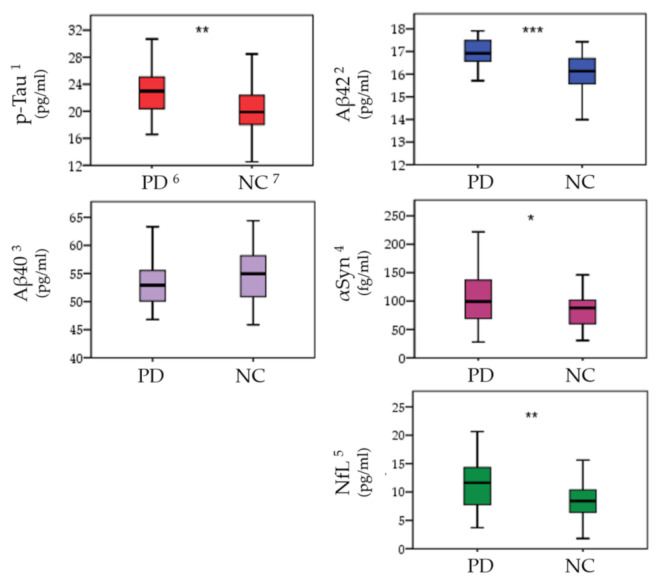
Analysis of plasma misfolded proteins levels between Parkinson’s patients (PD) and normal controls (NC). Misfolded protein levels are presented as mean (pg/mL or fg/mL) ± standard deviation. Plasma misfolded protein level data were compared with ANCOVA after controlling age and sex. ^1^ p-Tau (phosphorylated-Tau); ^2^ Aβ-42 (Amyloidβ-42); ^3^ Aβ-40 (Amyloidβ-40); ^4^ α-Syn (α-synuclein); ^5^ NfL (Neurofilament light chain); ^6^ PD (Parkinson’s disease); ^7^ NC (Normal control); * indicates Benjamini–Hochberg adjusted *p*-values < 0.05, ** *p* < 0.01, *** *p* < 0.001.

**Figure 2 jpm-11-00702-f002:**
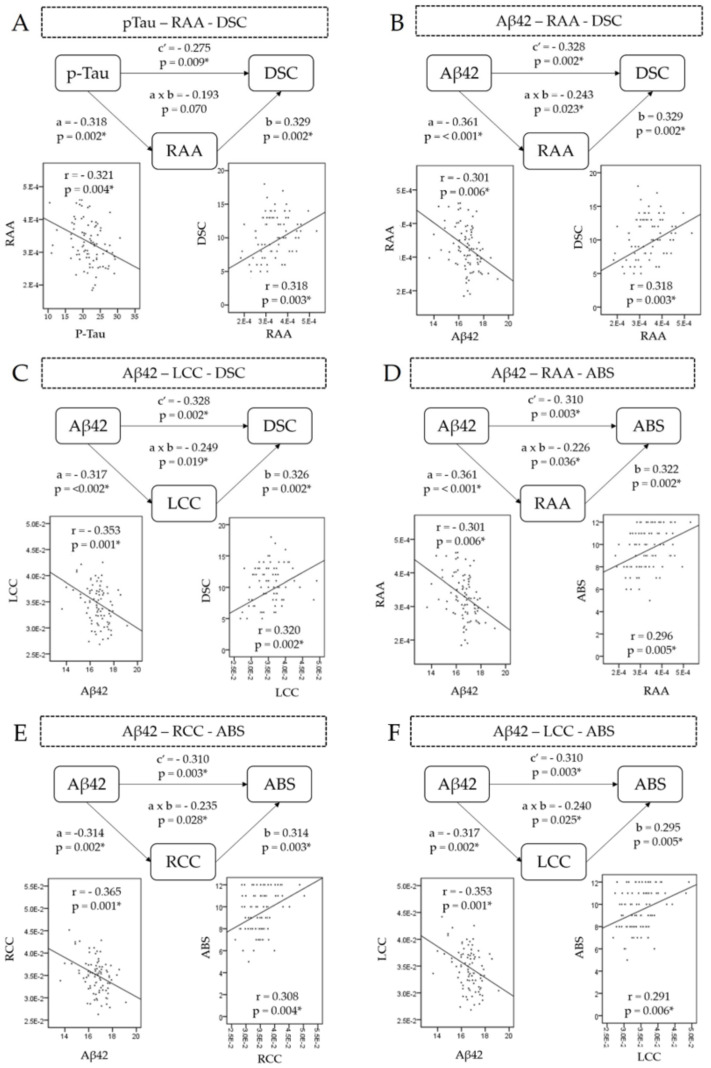
Mediation model. Mediation path diagram showing brain ROIs atrophy serve as a potential mediator between elevated plasma protein level and different cognitive functions. The correlation scatterplots indicate the relationship between misfolded protein level, brain atrophy, and cognitive functions. (**A**) revealed plasma p-Tau is negatively correlated with right accumbens area (r = −0.321, *p* = 0.004), and right accumbens area is positively correlated with digit symbol coding (r = 0.318, *p* = 0.003). Right accumbens area completely mediated p-Tau—RAA—DSC relationship. (**B**) revealed plasma Aβ42 is negatively correlated with right accumbens area (r = −0.301, *p* = 0.006), and right accumbens area is positively correlated with digit symbol coding (r = 0.318, *p* = 0.003). Right accumbens area partially mediated Aβ42—RAA—DSC relationship. (**C**) revealed plasma Aβ42 is negatively correlated with left cerebellar cortex (r = −0.353, *p* = 0.001), and left cerebellar cortex is positively correlated with digit symbol coding (r = 0.320, *p* = 0.002). Left cerebellar cortex partially mediated Aβ42—LCC—DSC relationship. (**D**) revealed plasma Aβ42 is negatively correlated with right accumbens area (r = −0.301, *p* = 0.006), and right accumbens area is positively correlated with abstract thinking (r = 0.296, *p* = 0.005). Right accumbens area partially mediated Aβ42—RAA—ABS relationship. (**E**) revealed plasma Aβ42 is negatively correlated with right cerebellar cortex (r = −0.365, *p* = 0.001), and right cerebellar cortex is positively correlated with abstract thinking (r = 0.308, *p* = 0.004). Right cerebellar cortex partially mediated Aβ42—RCC—ABS relationship. (**F**) revealed plasma Aβ42 is negatively correlated with left cerebellar cortex (r = −0.353, *p* = 0.001), and left cerebellar cortex is positively correlated with abstract thinking (r = 0.291, *p* = 0.006). Left cerebellar cortex partially mediated Aβ42—LCC—ABS relationship. * Indicates *p* < 0.05. Abbreviations: p-Tau (phosphorylated-Tau), RAA (right accumbens area), DSC (digit symbol coding), Aβ42 Beta-amyloid-42), LCC (left cerebellar cortex), ABS (abstract thinking), RCC (right cerebellar cortex), a (Coef_a_), b (Coef_b_), a × b (Coef_ab_), c′ (Coef_c′_).

**Figure 3 jpm-11-00702-f003:**
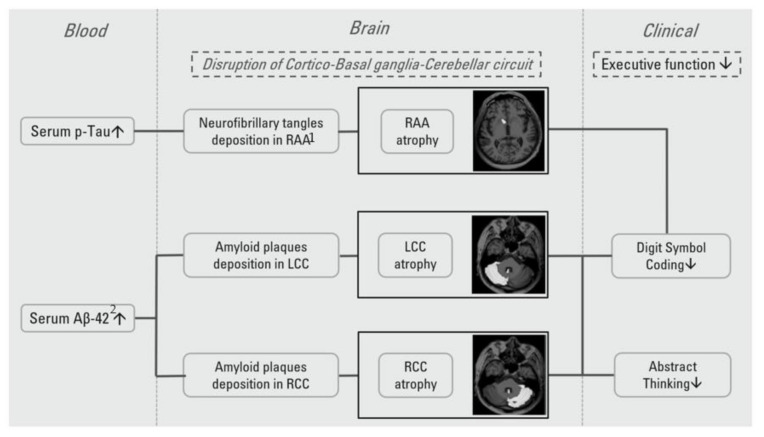
Misfolded deposition contributes to gray matter atrophy and cognitive dysfunction. We hypothesized that misfolded protein deposition in specific ROIs causes mediation effects. Plasma misfolded protein deposition is indicated and has a positive correlation with misfolded protein level in the brain. Deposition of p-Tau in RAA causes brain gray matter atrophy and is correlated with DSC and Aβ-42 in RAA and the bilateral cerebellum, mediated by executive function including ABS and DSC. This function includes the striato-thalamocortical circuit and cerebellum (cortico-basal ganglia-cerebellar circuit). Interconnections between cognitive regions of the basal ganglia, the cerebellum, and the cerebral cortex could provide part of the foundation for the executive control network. Misfolded protein accumulation in brain tissues with neuron toxicity leads to neuron death and causes gross brain structural changes such as gray matter atrophy, which are in turn correlated with clinical disease progression and cognitive function impairments. ^1^ RAA: right accumbens area, ^2^ Aβ-42: Beta-Amyloid-42.

**Table 1 jpm-11-00702-t001:** Basic demographic data. Data are presented as mean± standard deviation. Sex data were compared with Pearson chi-square test. Age data were analyzed with independent *t*-test. MMSE were compared using ANCOVA after controlling age, sex, and education level.

Clinical Demographics	PD ^1^ Patients (*n* = 54)	Controls (*n* = 37)	*p*
Age (year, mean ± SD)	60.69 ± 8.61	57.65 ± 8.02	0.093
Sex (M/F)	33/21	17/20	0.199
Education (year)	11.38 ± 4.35	13.78 ± 3.74	<0.001 *
Disease duration (year)	4.97 ± 3.32		
UPDRS ^2^ I	2.70 ± 2.11		
UPDRS II	11.63 ± 8.04		
UPDRS III	26.94 ± 17.53		
UPDRS 176	41.28 ± 25.99		
Modified H&Y ^3^	2.19 ± 1.22		
SE-ADL ^4^	65.02 ± 28.61		
MMSE ^5^	25.65 ± 3.97	29.16 ± 1.01	<0.001 *

^1^ PD (Parkinson’s disease); ^2^ UPDRS (Unified Parkinson’s disease rating scale); ^3^ Modified H&Y stage (Modified Hoehn and Yahr staging scale); ^4^ SE-ADL (Schwab and England activities of daily living scale); ^5^ MMSE (Mini mental state examination); * indicates *p*-value < 0.05.

**Table 2 jpm-11-00702-t002:** Analysis of neuropsychological assessments between Parkinson’s patients and normal controls. All the neuropsychological assessments were compared with ANCOVA after controlling age, sex, and education levels. Data are presented as mean± standard deviation.

Neuropsychological Assessments	PD ^2^ Patients (*n* = 54)	Controls (*n* = 37)	*p*
Attention			
Digit span	10.25 ± 2.78	11.54 ± 1.89	0.174
Attention	7.23 ± 0.73	7.73 ± 0.65	0.076
Orientation	17.04 ± 2.36	17.65 ± 0.86	0.570
Executive			
Digit symbol coding	7.15 ± 3.63	12.22 ± 2.38	<0.001 *
Arithmetic	8.88 ± 2.90	11.08 ± 3.18	0.008 *
Abstract Thinking	8.62 ± 2.06	10.62 ± 1.42	<0.001 *
Memory			
Short-term memory	9.30 ± 2.30	10.63 ± 2.28	0.163
Long-term memory	9.80 ± 1.07	10.00 ± 0.00	0.876
Information	9.88 ± 3.28	12.27 ± 3.20	0.012 *
Speech and Language			
Comprehension	9.71 ± 3.11	12.46 ± 3.05	0.001 *
Language	9.44 ± 1.06	9.84 ± 0.53	0.120
Semantic fluency	7.48 ± 2.39	8.54 ± 1.80	0.025 *
Visuospatial			
Picture completion	8.40 ± 3.26	11.08 ± 2.98	0.001 *
Block design	7.75 ± 3.43	11.03 ± 2.83	<0.001 *
Drawing	8.37 ± 2.51	9.84 ± 0.55	0.037 *
CASI ^1^	84.33 ± 16.00	93.70 ± 4.66	0.028 *

^1^ CASI (*Cognitive* Assessment Screening Instrument); * indicates *p*-value < 0.05. ^2^ PD (Parkinson’s disease).

**Table 3 jpm-11-00702-t003:** Correlation analysis of levels of plasma misfolded protein, neuropsychological assessments, and deep gray matter atrophy in PD group after controlling age, sex, and education level. The brain volumes were corrected by intracranial volumes.

PD Brain Regions										
	Cerebellar Cortex		Caudate Nucleus		Cerebellar Cortex		Caudate Nucleus		Accumbens ^1^	
**Hemisphere**	L ^2^		L		R ^3^		R		R	
	r	*p*	r	*p*	r	*p*	r	*p*	r	*p*
**Misfolded proteins**										
p-Tau	−0.311	0.005 *			−0.320	0.004 *	−0.342	0.002 *	−0.321	0.004 *
Aβ−42	−0.353	0.001 *	−0.397	<0.001 *	−0.365	0.001 *	−0.422	<0.001 *	−0.301	0.006 *
**Neuropsychological assessments**										
MMSE ^4^	0.413	<0.001 *	0.290	<0.001 *	0.362	<0.001*			0.409	<0.001 *
Arithmetic	0.416	<0.001 *			0.362	0.001*				
DSC ^5^	0.320	0.002 *							0.318	0.003 *
**ABS** ^6^	0.291	0.006 *			0.308	0.004 *			0.296	0.005 *

^1^ Accumbens (Accumbens area); ^2^ L (left), ^3^ R (right); ^4^ MMSE (mini mental state examination); ^5^ DSC (digit symbol coding); ^6^ ABS (abstract thinking). * Indicates Bonferroni adjusted *p* < 0.05.

**Table 4 jpm-11-00702-t004:** Potential brain mediators of plasma misfolded proteins on the cognitive functions. The corresponding statistical results of each path and mediation effect between the misfolded proteins (independent variable), cognitive functions (dependent variable), and brain areas (mediators) are described in terms of corresponding path coefficient, z value, and *p* value.

Clinical	Brain	Proteins	Path a			Path b			Path a × b			Path c′		
			Pcoef ^1^	z	*p*	Pcoef	z	*p*	Pcoef	z	*p*	Pcoef	z	*p*
**DSC** ^2^	**RAA ** ^ 4^	**p-Tau**	−0.318	−0.316	0.002	0.329	3.245	0.002	−0.193	−1.837	0.070	−0.275	−2.664	0.009
	**RAA**	**Aβ-42 ** ^ 5^	−0.361	−3.649	<0.001	0.329	3.245	0.002	−0.243	−2.310	0.023	−0.328	−3.237	0.002
	**LCC**	**Aβ-42**	−0.317	−3.155	0.002	0.326	3.217	0.002	−0.249	−2.394	0.019	−0.328	−3.237	0.002
**ABS** ^3^	**RAA**	**Aβ-42**	−0.361	−3.649	<0.001	0.322	3.170	0.002	−0.226	−2.127	0.036	−0.310	−3.041	0.003
	**RCC**	**Aβ-42**	−0.314	−3.125	0.002	0.314	3.090	0.003	−0.235	−2.243	0.028	−0.310	−3.041	0.003
	**LCC**	**Aβ-42**	−0.317	−3.155	0.002	0.295	2.878	0.005	−0.240	−2.278	0.025	−0.310	−3.041	0.003

^1^ Pcoef (path coefficient); ^2^ DSC (digit symbol coding); ^3^ ABS (abstract thinking); ^4^ RAA (right accumbens area); ^5^ Aβ-42 (Amyloidβ-42).

## Data Availability

Data available on request due to restrictions. The data presented in this study are available on request from the corresponding author. The data are not publicly available due to privacy and ethical.

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
