# Peer review of "Brain Atrophy Mediates the Relationship between Misfolded Proteins Deposition and Cognitive Impairment in Parkinson’s Disease"

_jpm, 2021, doi:10.3390/jpm11080702_

Round 1
Reviewer 1 Report
Yu et al studied the association between misfolded protein deposition, brain atrophy and cognitive impairment in 53 PD patients and 37 age matched healthy controls. Authors found a clear elevation of Plasma p-Tau and Abeta-42 in PD patients with RAA and bilateral cerebellum atrophy with significant lower scores on neuropsychological scores.
- The major problem with the study is authors did not present well the association between these parameters. Weak point of the study– there is no follow-up study involved and grading of PD.
- Authors need to present the plots/graph showing the association of imaging (brain atrophy), neuropsychological assessments and misfolded protein levels.
- Authors need to describe method clearly how they measured plasma biomarkers (kits used), need to plot the ROC (receiver operating characteristic) to show the sensitivity analysis of misfolded protein values for PD/control.
- Change the barograph to box plot. Bar graph scaling is not done well, it is difficult to see difference in misfolded protein level between case (PD) and control.
- Discussion needs to be accurate, author’s hypothesis on “Brain atrophy serves as a mediator of misfolded proteins on cognitive impairment” is not well judged. May be authors needs to change this part with accurate information that goes along with their findings.
- Authors need to subdivide the methods section
Study design, clinical scales
Neuroimaging assessments
Blood protein measurement
Statistical analysis and so on ……
Author Response
Dear Reviewer:
Thank you for providing us the opportunity to submit a revised manuscript titled "Brain atrophy mediates the relationship between misfolded proteins deposition and cognitive impairment in Parkinson's disease" to " Journal of Personalized Medicine". We appreciate the time and efforts that you have dedicated to providing the valuable feedback and comments on our manuscript. We have used the “Track Changes” function to highlight the revisions within the manuscript. The point-by-point responses according to your comments are in the attached file.

Reviewer 2 Report
In this manuscript, the author’s aims to investigate the association between structural abnormalities, plasma misfolded proteins and cognitive impairments. They found that PD patients experienced brain atrophy (bilateral cerebellar cortex, caudate nuclei, putamen, hippocampus, amygdala, RAA, and brainstem) as well as elevated misfolded proteins (p-Tau, Aβ-42, α-syn, NfL). From a neurological perspective, this is an interesting study that confirmed that plasma misfolded proteins (p-Tau and Aβ-42) are correlated with lower scores on neuropsychological assessments (MMSE, arithmetic, DSC and ABS) and brain atrophy (RAA, cerebellum and caudate nuclei), indicating the effects of misfolded proteins in neurotoxicity, neuron damage, and brain volume loss.
This manuscript is written in a concise and orderly manner. The methodologies are appropriate and aligned with the proposed objectives. The message from this manuscript is quite meaningful. I noticed only some spelling, minor errors, the lack of some definitions for abbreviations and an inconsistency regarding the number of patients studied (54 or 53 PD patients?).
Author Response
Dear Reviewer:
We are grateful for this opportunity to submit a revised manuscript titled "Brain atrophy mediates the relationship between misfolded proteins deposition and cognitive impairment in Parkinson's disease" to " Journal of Personalized Medicine". We appreciate the time and efforts that you have dedicated to providing the valuable feedback and comments on our manuscript. We have used the “Track Changes” function to highlight the revisions within the manuscript. The response according to your comments is in the attached file.

Round 2
Reviewer 1 Report
Authors addressed all the comments/changes that I suggested. Manuscript meeting the standards of the journal with these changes.
I agree with the authors choices of figure placement. Go a head with the changes that authors suggested related to figures (moving to supplement/placing in main figures).